# Learning Visuo-Haptic Skewering Strategies for Robot-Assisted Feeding

**Priya Sundaresan**
Stanford University
priyasun@stanford.edu

**Suneel Belkhale**
Stanford University
belkhale@stanford.edu

**Dorsa Sadigh**
Stanford University
dorsa@cs.stanford.edu

**Abstract:** Acquiring food items with a fork poses an immense challenge to a robot-assisted feeding system, due to the wide range of material properties and visual appearances present across food groups. Deformable foods necessitate different skewering strategies than firm ones, but inferring such characteristics for several previously unseen items on a plate remains nontrivial. Our key insight is to leverage visual and haptic observations during interaction with an item to rapidly and reactively plan skewering motions. We learn a generalizable, multimodal representation for a food item from raw sensory inputs which informs the optimal skewering strategy. Given this representation, we propose a zero-shot framework to sense visuo-haptic properties of a previously unseen item and reactively skewer it, all within a single interaction. Real-robot experiments with foods of varying levels of visual and textural diversity demonstrate that our multimodal policy outperforms baselines which do not exploit both visual and haptic cues or do not reactively plan. Across 6 plates of different food items, our proposed framework achieves 71% success over 69 skewering attempts total. Supplementary material, datasets, code, and videos can be found on our website.

**Keywords:** Assistive Feeding, Deformable Manipulation, Multisensory Learning

## 1 Introduction

Realizing the full capabilities of assistive robots in the home, hospitals, or elderly care facilities remains challenging due to the dexterity required to complete many day-to-day tasks. Eating free-form meals is one such example with many nuances in perception and manipulation that can be easy to overlook. However, automating the task of feeding, one of six essential activites of daily life (ADL) [1], has the potential to improve quality of life for over one million people in the U.S. who are unable to feed themselves due to upper-extremity mobility impairment, their caregivers, families with young children and elders, and anyone impacted by the substantial time and effort required in meal preparation and feeding [2, 3, 4].

In recent years, there have been significant efforts to tackle the challenging problem of robot-assisted feeding. Solutions on the market have limited traction as they rely heavily on pre-programmed trajectories, pre-specified foods, have limited autonomy, or require manual utensil interchange [5, 6]. Meanwhile, academic research on assistive feeding largely centers around data-driven methods but has yet to show widespread generalization across food groups [7, 8, 9, 10]. As a necessary first step towards feeding, we focus on the problem of *bite acquisition* — acquiring bite-sized items from a plate or bowl — using a robot with a fork-equipped end-effector. Developing a bite acquisition strategy sensitive to differences in *geometry* and *deformation* both *across and within* food classes is a challenging problem: Skewering position and orientation matters for items with irregular shape, such as a broccoli floret where skewering at the stem is preferable to the head for stability of the acquisition. The fragility of food also affects the optimal skewering strategy, as delicate items such as thin banana slices are more likely to slip off a fork oriented vertically and instead benefit from an angled fork insertion and scooping strategy [11]. On the other hand, hard baby carrots require a vertical insertion angle for effective and stable acquisition [11]. In addition, instances within the same class of foods can also exhibit visual similarity but textural contrast (raw vs. boiled carrots, silken vs. extra firm tofu, cheddar vs. mozzarella cheese); choosing the optimal skewering strategy therefore depends on more than just vision.

Prior works show that food classification objectives can lead to visual features that can be used for downstream policy learning [7, 8, 9]. They also introduce an action taxonomy for skewering to

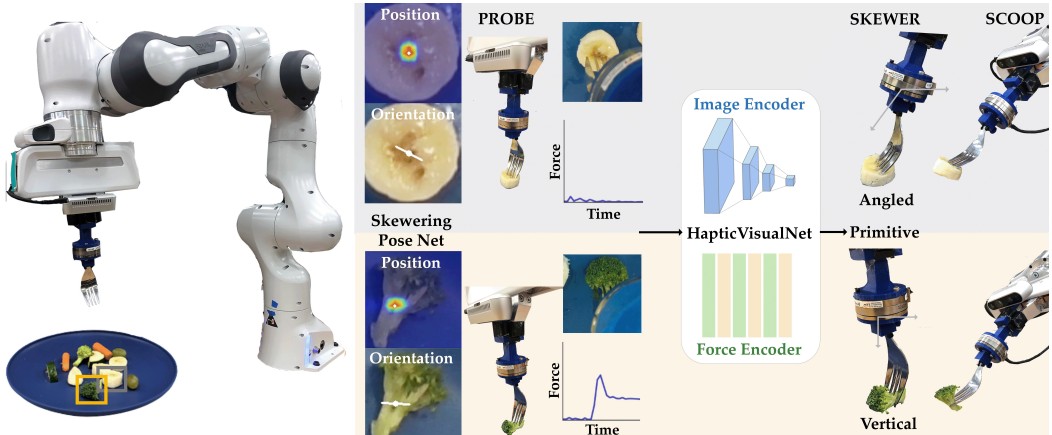

Figure 1: *Left:* Our method learns zero-shot skewering of food items with a Franka Panda robot. Given an overhead plate observation, we localize food items and *probe* them to reveal haptic and visual data. Using the multimodal data as input, HapticVisualNet infers the optimal skewering trajectory on the fly – *angled skewering* for soft items like banana slices, or *vertical skewering* for firm textures such as a broccoli stem.

discretize the complex space of possible acquisition trajectories [7, 8, 9, 12, 13]. Although these visual-only skewering strategies are able to classify food with different *geometry*, they lack critical information about *deformation* and may fail to differentiate between foods *within* the same food class that appear similar but have drastically different properties, such as boiled and raw carrots.

Preliminary experiments suggest that a boiled carrot requires the fork to skewer the item at an angle to avoid breakage or dropping during acquisition, whereas a rigid carrot requires a more forceful vertical approach to pierce the item. Gordon et al. [10] try to address this issue by leveraging post-hoc haptic feedback to update a visual policy after skewering. This work requires multiple trials of interaction per unseen food item to reason about item *deformation* through haptic feedback. However, repeated skewering attempts can easily damage fragile items (e.g. overcooked carrots or thin slices of banana) and potentially change the properties of the food, leading to breakage or squishing over multiple robot interactions. These repeated interaction strategies do not scale to unseen food *classes* either, for the same reasons. On the other hand, open-loop strategies that do not adapt skewering plans mid-motion are also limited in their ability to handle unseen items with unknown properties.

Therefore, bite acquisition methods should be able to *zero-shot* generalize to new foods both within and across food classes, just like how humans skewer bites of food without the need for multiple interactions with the food.

Our key insight is to jointly fuse haptic and visual information during a single skewering interaction to learn a more robust and generalizable food item representation. We develop a bite acquisition system along with a visuo-haptic skewering policy that leverages this learned representation. The proposed representation informs the skewering policy of both *geometry* and *deformation* food properties on-the-fly, thus enabling a reactive policy which zero-shot generalizes to unseen plates of food *within and across* food classes. Our experiments with a wide range of food items with varying geometry and deformability demonstrate that our method outperforms those that (1) do not jointly use haptic and visual cues and (2) do not reactively plan upon contact, achieving $71\%$ skewering success across 21 items total. Our contributions include:

- A skewering system that employs coarse-to-fine visual servoing to approach a food item, sense multimodal properties upon contact, and reactively plan skewering in the same continous interaction
- A zero-shot skewering policy that captures geometry and deformation by fusing visual and haptic information with demonstrated generalization to unseen food items
- Experimental validation on diverse seen and unseen food items, with varying degrees of visual likeness and deformation
- An open-source dataset for multimodal food perception and custom end-effector mount designs, which we hope expands the scope of assistive feeding research

## 2 Related Work

We build on prior works studying multisensory robot learning both within and beyond the food domain. In this section, we will discuss related work in robot-assisted feeding, food manipulation, and more generally multimodal robotic perception and manipulation.

**Robot-Assisted Feeding**    Feeding can be split into two stages: bite acquisition and bite transfer. Previous work in bite transfer — transferring an acquired bite to a user's mouth for consumption — suggests that transfer is largely contingent upon acquisition [14, 7, 13]. To enable reliable bite acquisition and thus transfer, recent acquisition frameworks combine image-based perception [7, 8, 9, 10, 13] — bounding boxes and food pose estimates — with an action space consisting of parameterized primitives that modulate fork roll/pitch relative to item geometry. SPANet (Skewering Position Action Network) [8] is one such forward model mapping food image observations to actions, which has been shown to reasonably clear plates containing 16 types of fruits and vegetables. SPANet is trained on 2.5K fork interactions (81 hours of supervision [9, 8]) which does not readily scale to new foods. Follow-up works aim to rapidly adapt SPANet to unseen food items using a contextual bandit to learn the optimal primitive selection strategy from real interactions. Approaches include updating SPANet predictions online by observing the binary outcome of acquisition attempts on unseen items [9], and additionally haptic time-series readings recorded post-hoc [10]. A key assumption in [10] is that the visual context observed *pre-skewer* and the haptic context observed *post-hoc* are equivalent alternate representations for the underlying food state, which does not always hold (e.g., firm and soft tofu appear to be almost identical but yield different haptic readings). In our work, we do not restrict ourselves to this assumption and consider the more general setting where visually similar items may have different physical properties. In addition, we do not assume access to repeated interaction trials with the food, and consider a zero-shot planning setting. Bhattacharjee et al. [12], Song et al. [15] explore classifying food compliance or skewering outcomes from haptic data from a single interaction, but delegate reactive planning given these representations to future work. To address these gaps, we learn a multisensory policy that learns zero-shot skewering from *pre-skewer* and *post-contact* paired images and haptic readings.

**Food Manipulation**    Recent simulated benchmarks for household food manipulation explore food preparation, lunch packing, and food storage [16]; pouring water [17]; pile manipulation for chopped food [18]; and drinking/feeding [19]. While these works largely abstract away the state space of food, recent work in real robotic food slicing explores multimodal food representation learning from interaction using datasets consisting of paired visual, tactile, and audio information [20, 21, 22, 23, 24]. In particular, Gemici and Saxena [20] propose to infer haptic properties ('brittleness', 'tensile strength', 'plasticity', etc.) from probing actions that can inform slicing actions, but does not consider visual properties. Likewise, Zhang et al. [25] use vibrations from probing interactions to adapt slicing motions. The learned haptic representations from these work are not directly transferable to acquisition due to hardware and task differences, but we adopt the notion of using probing actions to inform skewering in our work.

**Integrating Vision and Haptics in Robotics**    Vision-only manipulation has proven effective in robotic domains such as semantic grasping [26, 27] and deformable manipulation [28], but contact-rich tasks such as peg insertion [29] or robotic Jenga [30] have been shown to benefit from combining vision, force, and proprioception as inputs to a learned policy [31, 32]. In light of these works, we propose to learn reactive, visuo-haptic policies for bite acquisition which remains largely unexplored.

## 3 Method

Our goal is to learn a multisensory manipulation policy which outputs skewering actions to clear a plate of bite-sized food without any previous skewering attempts for this food item. We consider foods of varying degrees of geometric, visual, and textural similarity, all on the same plate. We first formalize the bite acquisition setting in Section 3.1 and introduce our action space to tackle the problem in Section 3.2. Next, we discuss our interaction protocol for sensing visuo-haptic properties of food (Section 3.3), enabling us to learn a multimodal skewering policy (Sections 3.4-3.5).

### 3.1 Problem Formulation

At each timestep $t \in 1, \ldots, T$, we assume access to a the current RGB-D image observation $I_t \in \mathbb{R}^{W \times H \times C}$ of a plate of food and an $N$-length history of haptic readings $H_t \in \mathbb{R}^{N \times 6}$, denoting 6-axis readings from an F/T sensor on a fork-mounted end-effector. Similar to prior work, we consider an action space parameterized by fork $(x, y, z)$ position, roll $\gamma$, and pitch $\beta$. We define an action $a_t \in \mathcal{A}$, visualized in Fig. 1, as follows [9, 10]:

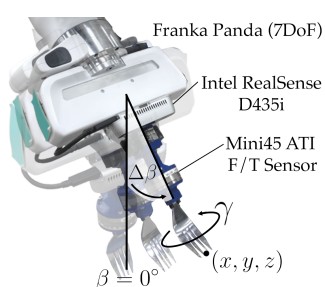

$$a_t = (x, y, z, \Delta z, \gamma, \Delta \beta) \qquad (1)$$

Figure 2: Skewering action space: fork pitch $\beta$, roll $\gamma$, and position $(x, y, z)$.

At time $t$, the fork starts in position $(x, y, z)$ with pitch $\beta = 0°$ and roll $\gamma$ and moves downward $\Delta z < 0$ while optionally tilting $\Delta \beta \geq 0°$ to skewer an item (Figure 2).

We use $l_t[a_t] \in \{0, 1\}$ to denote the binary loss of executing action $a_t$, where $l_t[a_t] = 1$ denotes failure, for example food failing to be picked up, slipping off the fork after skewering, or breakage or damage as a result of skewering. We opt for a binary loss objective as it is difficult to quantify more nuanced notions of success such as degree of damage to an item or stability of a skewer.

We aim to learn a policy $\pi_\theta(a_t \mid I_t, H_t)$ that minimizes $\sum_t^T l_t[a_t]$ given no previous interactions. Our policy learns to map visual and haptic information to a discrete set of skewering primitives using a small but diverse labelled dataset that can be collected offline. By conditioning the policy on haptic readings during a short initial contact period with the the food item, our method HapticVisualNet can extract food properties that are inaccessible to vision using only a single skewering interaction. First we outline how we represent the discrete set of primitives for skewering.

## 3.2 Skewering Action Primitive Parameterization

To successfully skewer a food item, the fork position and roll must align with the location and orientation of a food item. The fork pitch must also adapt to the compliance of a food item (i.e. a soft banana slice favors an *angled* fork approach to prevent slip, while a raw carrot favors a *vertical* approach for piercing). Thus, our action space employs two primitives, `vertical skewer` or `angled skewer`, to account for rigid or compliant items, respectively. We implement `vertical skewer` with $\Delta \beta = 0°$, denoting no tilt during skewering, and `angled skewer` with $\Delta \beta > 0°$, where the fork gradually tilts from vertical to an angled approach during insertion into a fragile food item (Figure 2). Prior work includes both angled and vertical skewering strategies amongst an even larger set of primitives, but we empirically find that our simplified taxonomy reduces redundancy in this larger action space and can handle an equivalently broad range of food items, evaluated in Section 4 [8, 10]. In order to decide between these strategies on-the-fly, we condition our policy not only on visual information but also on haptic information at the point of contact with the food item, discussed in the next section.

## 3.3 Sensing Multimodal Data Via Probing

We introduce a *probing* motion to obtain visuo-haptic information about a food item by bringing the fork in contact with the item surface but without actually skewering it. The multisensory information collected from probing serves as input to $\pi_\theta$ which rapidly decides the skewering primitive to execute. Between the probing and skewering phases, the fork remains stationary and in constant contact with the food item, thus enabling a fluid transition between phases.

Our *probe-then-skewer* approach requires localizing an item and approaching with precision so as to not accidentally shift or topple it while making contact. To accomplish this, we first detect items from a plate image using a pre-trained RetinaNet food bounding box detector from Gallenberger et al. [7]. Similar to prior work, we also train a network (SkeweringPoseNet) which refines the estimated item location by predicting a keypoint for the item center within the local bounding box, and a fork roll angle $\hat{\gamma}$ with which to approach [8]. We can obtain the 3D predicted item location $(\hat{x}, \hat{y}, \hat{z})$ in robot frame by using depth information. Next, we continuously servo to the item using a learned model (ServoNet) which detects the fork-item offset as keypoints from streamed RGB images. Using this framework, we probe a food item and record a post-contact image observation $I_t$ and the short window of force magnitude readings $H_t$. In Section 3.4, we discuss how these multisensory readings inform the optimal skewering strategy.

## 3.4 Multimodal Representation and Policy Learning

To learn $\pi_\theta(a_t \mid I_t, H_t)$, we propose HapticVisualNet, a network which takes observations captured by the probing motion and reactively outputs the appropriate skewering action. HapticVisualNet takes as input a post-contact image of a food item $I_t \in \mathbb{R}^{W \times H \times 3}$ and $H_t$, the force magnitude readings recorded from the F/T sensor during the first $N$ milliseconds of contact in the initial probing period.

HapticVisualNet maps $(I_t, H_t)$ to an $|\mathcal{A}|$-d vector denoting the likelihood of success for each action primitive, in our case `vertical skewer` and `angled skewer`. The model first encodes visual information and haptic information separately, and then concatenates these features to produce a joint visuo-haptic representation. The policy then predicts action success likelihood from this representation. We implement HapticVisualNet as a multi-headed network with a ResNet-18 backbone for the visual encoder and a LSTM for the haptic encoder. We pass the concatenated visual and haptic encodings to a linear layer followed by a softmax to obtain primitive successes, and choose the maximum likelihood predicted primitive as the skewering action.

## 3.5 Training and Data Collection

We train HapticVisualNet on a small but diverse dataset of 300 paired post-contact images and haptic readings, augmented 8x using image affine and colorspace transforms as well as temporal scaling and shifting of the haptic readings. The dataset consists of *hard* items labeled `vertical skewer` (raw carrots / broccoli / zucchini / butternut squash, grapes, cheddar cheese, and celery) and *soft* items labeled `angled skewer` (banana, kiwi, ripe mango, boiled carrots / broccoli / zucchini / butternut squash, avocado, and mozzarella cheese).

In practice, we use an $N = 26$ ms. contact window of haptic readings, from the initial probing period, which we find adequately captures force-surface interactions. Using a 20-30 ms. window of contact is also a common choice of haptic representation in other reactive, contact-rich manipulation settings [29, 10]. For each paired example, we manually assign the optimal primitive label — `vertical` or `angled skewer` — based on whether the annotator considers the item hard or soft. This process requires 3 hours of data collection and labeling time total (a 27x reduction from the 81 hours reported in SPANet [10]), and *without* the need for actual skewering attempts during data collection. We intend for the inclusion of haptic data to prevent overfitting to visual features, enabling a more generalizable, food representation trained with greater sample efficiency. Both ServoNet and SkeweringPoseNet are implemented using a ResNet-18 backbone, each trained on 200 images of the same items (2 hours of supervision) and augmented to a dataset of 3,500 paired images and annotations. Additional training and implementation details are in Appendix C.

In summary, our method leverages offline datasets to learn visuo-haptic features to more robustly predict between a set of discrete skewering actions. The probing motion used to obtain haptic information is connected seamlessly to the chosen skewering motion, leading to one continuous and adaptive zero-shot skewering policy aware of haptics *and* vision.

## 4 Evaluation

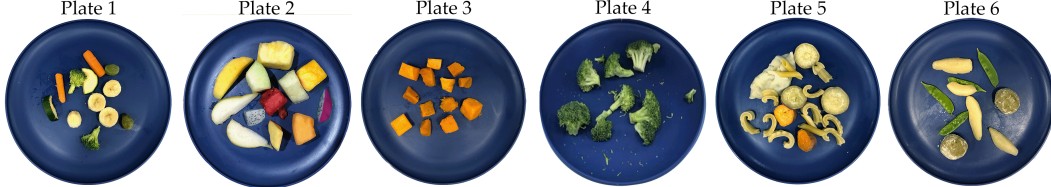

Figure 3: 6 Plates for Evaluation, covering a wide range of foods. From left to right: **1**: Raw banana, broccoli, zucchini, carrot, grapes, cucumber. **2 (Unseen)**: Raw pineapple, mango, dragonfruit, canteloupe, honeydew, pear. **3**: Raw butternut squash, boiled butternut squash. **4**: Raw broccoli florets. **5 (Unseen)**: Pasta, dumpling, boiled yam/sweet potato, raw yam/sweet potato. **6 (Unseen)**: Ice cream mochi, snow peas, canned peaches.

In this section, we seek to evaluate (1) the benefits of combining both vision and haptics for bite acquisition as opposed to only using a subset of modalities, (2) the effectiveness of reactive skewering compared to open-loop strategies, and (3) the generalization capabilities of our system to previously unseen foods. We first perform classification ablations of HapticVisualNet in (Section 4.1) and then deploy our system in the real world for trials on both seen and unseen foods (Section 4.2-4.4).

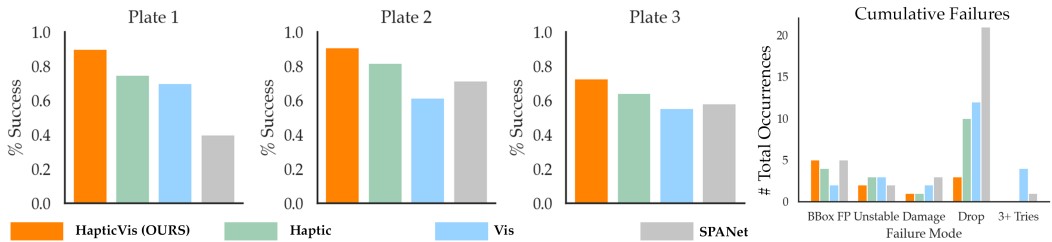

Figure 4: **Skewering Success and Failure Modes**: We visualize the number of items acquired over total acquisition attempts for all methods across Plates 1-3. Failure modes include exceeding the maximum number of consecutive attempts (3+ tries per item), dropping after skewering, the item being unstable on the fork after skewering (affecting transfer), damage or breakage to fragile items, or failure to detect an item due to bounding box anomalies. HapticVisNet (ours) performs best on each plate, while causing the least failures.

## 4.1 Ablative Studies

We evaluate the contributions of both visual and haptic data towards primitive classification by training and evaluating HapticVisualNet against two variants which observe exclusively the post-contact haptic readings or post-contact image after probing.

Figure 5 shows the confusion matrices for classification accuracy. HapticVisualNet achieves the highest overall and per-class classification accuracy, and omitting either modality hurts accuracy. We hypothesize that the reduced performance of the vision-only model stems from visually similar but texturally dissimilar foods in our dataset, for which inferring the primitive is challenging without haptic context. On the other hand, the haptic-only model learns a naïve solution of mapping high magnitude contact events to

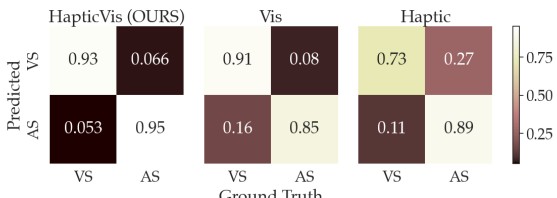

Figure 5: Confusion matrices for classification accuracy for the skewering primitive (`angled skewer` or `vertical skewer`), on a held-out test dataset of 60 images, for each model. Darker off-diagonals and lighter on-diagonals indicate more accurate models.

`vertical skewering` and low readings to `angled skewer`. This is brittle for anomalous foods like broccoli which benefit from a `vertical skewer`, yet may yield low contact readings if the fork comes in contact at the head instead of the stem. Similarly, the fork can easily penetrate a thin banana slice and touch the plate during probing, yielding high contact readings when an `angled skewer` is still preferable. In the subsequent sections, we evaluate HapticVisualNet (HapticVis) against the haptic only (Haptic), vision only (Vis), and SPANet baselines on real food acquisition trials. See Appendix D for additional ablations of the learned multimodal representation and HapticVisualNet sample efficiency.

## 4.2 Hardware Setup

Our setup consists of a 7DoF Franka Emika Panda robot with the default gripper. We outfit the gripper with a custom 3D-printed mount comprised of a standard fork, a Mini45 ATI F/T sensor, and a D435i RealSense camera. We perform all acquisition trials on a plastic dinner plate on an anti-slip surface. We instantiate each primitive, parameterized according to Equation (1) as follows, assuming a fixed $\Delta z$ and discretized $\Delta \beta$:

- `probe` $= (\hat{x}, \hat{y}, \hat{z}\text{-}\texttt{APPROACH\_HEIGHT}, \text{-}\texttt{APPROACH\_HEIGHT}, \hat{\gamma}, 0°)$
- `vertical skewer` $= (\hat{x}, \hat{y}, \hat{z}, \text{-}\texttt{DT*0.17m/sec}, \hat{\gamma}, 0°)$
- `angled skewer` $= (\hat{x}, \hat{y}, \hat{z}, \text{-}\texttt{DT*0.08m/sec}, \hat{\gamma}, 65°)$

Here, $(\hat{x}, \hat{y}, \hat{z})$ denotes a predicted food item location, obtained by deprojecting a predicted pixel in a depth image from ServoNet to a 3D location. SkeweringPoseNet also predicts the fork roll $\hat{\gamma}$. We first `probe` the item starting from an `APPROACH_HEIGHT` of 0.01cm, observe a post-contact image $I_t$, and record haptic readings $H_t \in \mathbb{R}^{26}$ over the first 26-milliseconds of contact. Given these inputs, HapticVisualNet infers either `vertical skewer` or `angled skewer` which we execute.

The robot controller runs at 20Hz (`DT = 0.05`), and primitives terminate early if the fork reaches a pre-defined $z$-distance (the plate height) or force limit. After skewering, the end-effector *scoops* until the fork is nearly horizontal with $\beta = 80°$, emulating the start of a feasible *transfer* trajectory.

### 4.3 Baselines

We deploy all methods — HapticVis (ours), Haptic, Vis, and SPANet — on the real robot setup of Section 4.2 and classification networks trained according to Section 4.1. The HapticVis, Haptic, and Vis methods all perform *probing-then-skewering*, but run inference using both, only haptic, or only visual sensory information obtained from probing, respectively.

We also implement SPANet given the pre-trained visual models and original taxonomy of six skewering primitives reported in [8]. SPANet performs zero-shot primitive inference given an overhead image observation of a food item, without probing. SPANet still uses our ServoNet to plan, analogous to the original implementation which similarly accounted for fork-food precision error.

### 4.4 Real World Bite Acquisition Results

We compare all methods on the challenging task of clearing plates containing 10 bite-sized food items, evaluated according to skewering success and the distribution of skewering failure modes encountered (Table 1). We define a skewering success as one in which the fork picks up the item with at least 2 tines inserted and the item remains on the fork for up to 5 seconds after scooping as in [8]. Failure modes are detailed in Figure (4). Between successful acquisitions, a human operator removes the acquired item from the fork. Upon skewering failure, an attempted item remains on the plate and can be re-attempted up to 3 times before being manually removed and marked as a failure.

**Plates 1-3 – Full System Evaluation**: We first perform a full-system evaluation of HapticVis and all baselines on 3 plates. Plate 1 contains in-distribution fruits and vegetables that HapticVisualNet was trained on which include both textural and visual diversity, Plate 2 consists of unseen assorted fruits with visual diversity but similar textures, and Plate 3 contains in-distribution boiled and raw butternut squash cubes which appear similar but greatly differ in softness. When adjusted for perception failures (e.g., bounding box false negatives) which affect all methods, HapticVis outperforms all methods across Plates 1-3 (Figure 4). The bulk of HapticVis failures center around near-misses or perception failures which are less drastic than damaging items or exceeding skewering attempts (Figure 4). Vis slightly underperforms Haptic across all plates and achieves lowest performance on Plate 3, where mispredicted `vertical skewer` or `angled skewer` actions can miss/damage soft-boiled squash, or fail to penetrate hard raw squash. SPANet achieves lowest performance, mostly due to erroneously executing vertical strategies to pick up soft items like bananas and ripe dragonfruit in many cases. SPANet's underperformance relative to HapticVis, Haptic, and Vis suggests the effectiveness of reactive strategies compared to open-loop acquisition, and that our simplified action space is just as expressive as SPANet's larger taxonomy. We acknowledge that the performance gap may in part be attributed to hardware differences (different robot, different fork mount) in the original SPANet compared to our re-implementation.

**Plate 4 – Texturally Misleading Food**: While Haptic is the most competitive baseline on Plates 1-3, we run additional experiments between HapticVis and Haptic on a plate of only broccoli florets (Plate 4). In cases where the stem is occluded from view or the fork servos to the leafy region to make contact, Haptic tends towards misclassifying the low readings as `angled skewer`, leading to frequent failures to pierce the item which HapticVis is better equipped to recognize and avoid.

**Plates 5-6 – Generalization to Out-of-Distribution Foods**: Finally, we stress-test the generalization capabilities of HapticVis on two plates of unseen foods, Plate 5 and Plate 6, containing the items listed in Table 1. Across Plates 5-6, HapticVis achieves 58% success, a 19% improvement over Vis, indicating the promise of multimodal representations for zero-shot food skewering. HapticVis performs best on soft canned pears and boiled/raw root vegetables which are most comparable to the items in the training distribution, closely followed by pasta and ice cream mochi. The majority of failures occur due to the fragility of dumplings and thinness of snow peas which are difficult to pierce. Still, by fusing haptic and visual information, HapticVis is better equipped to generalize to visually and texturally diverse foods. See Appendix E for additional stress-tests of HapticVisualNet.

Overall, HapticVisualNet benefits from the use of both vision and haptics from just a single interaction, and outperforms single-modality baselines for a variety of challenging food plates. We show that skewering strategies that reactively update the strategy (HapticVis, Haptic, Vis) are more robust to texturally and visually diverse food items than open loop strategies (SPANet). With its multimodal representation, HapticVisualNet can also zero-shot generalize to challenging unseen food classes.

| | Plate Type | | | | # Items Acquired / Total Attempts | | | |
|---|---|---|---|---|---|---|---|---|
| *Plate* | *Items* | *Visuals* | *Haptics* | *Category* | *HapticVis* | *Haptic* | *Vis* | *SPANet* |
| 1 | Assorted fruits and vegetables | Diverse | Diverse | Seen | **9/10** | 9/12 | 7/10 | 8/20 |
| 2 | Assorted tropical fruits | Diverse | Similar | Unseen | **10/11** | 9/11 | 8/13 | 10/14 |
| 3 | Boiled/raw butternut squash | Similar | Diverse | Seen | **8/11** | 9/14 | 10/18 | 7/12 |
| 4 | Broccoli florets | Similar | Similar | Seen | **8/13** | 7/17 | – | – |
| 5 | Pasta, dumplings, boiled/raw root veggies | Diverse | Diverse | Unseen | 7/11 | – | **9/13** | – |
| 6 | Mochi, snow peas, canned pear | Diverse | Diverse | Unseen | **7/13** | – | 5/23 | – |
| | | | | | **71%** | 63% | 51% | 54% |

Table 1: **Physical Results:** We evaluate the ability of all methods to clear 6 plates (Figures 4), each initially containing 10 items. The number of items acquired refers to items successfully skewered ($\leq 10$ in all cases due to dropped items or early termination with bounding box false negatives). The total attempts refers to all attempted acquisitions until termination or clearance ($\geq 10$ in all cases due to failed items remaining on the plate for up to 3 consecutive re-attempts). HapticVis outperforms baselines in 5/6 plates. The symbol – denotes experiments that are not useful in the specific testing scenario.

## 5 Discussion

**Summary** In this work, we present a framework for zero-shot food acquisition of a diverse range of bite-sized items using multimodal representation learning. Our approach uses interactive probing to sense complex, multisensory food properties upon contact in order to reactively plan a skewering strategy. We deploy the learned policy along with a learned visual servoing controller on a robot for zero-shot skewering of unseen food items. Our experiments span a wide range of food appearances and textures, and validate the need for multimodal reasoning and reactive planning to clear plates.

**Limitations and Future Work** One limitation in our approach is the use of a small action space for acquisition. While our set of primitives can generalize to a wide range of foods, our current strategies are not equipped to handle thin, flat items like finely sliced produce or leafy salad greens, which may require new techniques like positioning the fork under an item to scoop, gathering multiple items together before skewering, or using plate walls for stabilization. Other food groups like noodles would benefit twirling, and foods with irregular geometries might require toppling into a stable pose before skewering. Another limitation in this work is the supervision used to currently train our multimodal policy. However, since our policy learns from sparse labels, we are excited by the possibility of automatically detecting skewering outcomes to self-supervise the training procedure for HapticVisualNet. Finally, in future work we hope to tackle challenging food groups such as filled dumplings which easily break and extremely thin snow peas which are hard to pierce.

## Acknowledgments

This work is supported by NSF Awards 2132847, 2006388, and 1941722 and the Office of Naval Research (ONR). Any opinions, findings, and conclusions or recommendations expressed in this material are those of the author(s) and do not necessarily reflect the views of the sponsors. Priya Sundaresan is supported by an NSF GRFP. We thank our colleagues for the helpful discussions and feedback, especially Tapomayukh Bhattacharjee, Jennifer Grannen, Lorenzo Shaikewitz, and Yilin Wu.

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
