# OpenReview forum: "Learning Visuo-Haptic Skewering Strategies for Robot-Assisted Feeding"
_robot-learning.org/CoRL/2022/Conference — CoRL 2022 Oral_

### Official Review · Reviewer_b6eX · 2022-07-30

**Originality:** Good
**Technical Quality:** Good
**Clarity Of Presentation:** Very Good
**Impact:** 3

**Recommendation:**

Weak Accept: I recommend accepting the paper, but will not argue for my recommendation if the majority of other reviewers have a different opinion.

**Summary:**

The paper proposes a robotic system for skewering pieces of food from a plate. Taking visual and force data as input, a learned policy controls the robot to (1) detect pieces of food, (2) move the fork via visual servoing, (3) probe the food, (4) and select a primitive based on the measured probing data. The key contribution is the NN mapping the visual and haptic probing data to one of the two primitive types; they differ in the fork pitch angle intended for either hard or soft objects. The data (300 probing examples) was human annotated. Experimental evaluation was done on various foods and compares the approach with different input modalities as well as one baseline.

**Issues:**

We suggest to address following comments (in no particular order):

After playing around with various foods, we think that both primitives could be combined into a vertical skewer motion and a subsequent angled lifting with support from the plate. In this regard, we suggest (similar to the paragraph above) adding some ablative experiments about the action space.

The action space definition could be made clearer. We suggest including only the learned action space in eq. (1), and clarify that z is via depth image, delta z is fixed, delta beta are two discrete values only.

Similarly, the reward definition seems unusual. How many timesteps does an episode have? Isn’t the episode stopped after failing, so the loss would always sum up to one? Given the description, the system seems to actually learn a binary reward from the human annotation if the whole skewing primitive is successful.

Code and dataset were not published during the time of review, so please do so now or remove their references from the paper. Of course, feel free to add them to the website anytime.

Why weren't the internal force sensors used? Moreover, we suggest giving a brief (one-sentence) overview in 4.2 about the used control software for the robot.

Why is the ServoNet necessary? In general, the absolute position accuracy of the Franka robot is below 1 mm so it should be able to move to a target position without servoing.

We suggest replacing several arxiv citations with their conference/journal counterparts.

In line 222, ”only” is doubled.


**Quality Of The Limitations Section:**

Additional details required

**Reviewer Expertise:**

4: The reviewer is confident but not absolutely certain that the evaluation is correct

**Robotics Focus:**

Sufficient demonstration on hardware

**Strengths And Weaknesses:**

The paper is nicely written and easy to follow. We like the application as well as the idea of probing the food to get very simple closed loop feedback. Its implementation, as seen in the video, seems fast and nearly unnoticeable. Moreover, the experiments study generalization extensively with a wide range of foods. While we think that the contribution itself is not particularly large, the sound implementation and the task might be interesting for the community.

However, we think that the quantitative results (e.g. a 71% success rate) aren’t very strong. Therefore, we would like to see a more detailed ablative study. In particular, 300 annotations (in 3h of data collection) seems very few for a data-driven approach. How do the results improve with training data? Broccoli florets (plate 4) have a 61% success rate, despite being a trained food. Is this a matter of training data? What are the success rates for using only one primitive?

**Summary Of Recommendation:**

We think that the originality of this work is incremental, as probing is a common solution in (robotic) measurement tasks. Moreover, the general ideas are only applicable to the task of skewering foods from plates.

---

> ### Author Response · Authors · 2022-08-22
> **Response to Reviewer b6eX [1/2]**
>
> Thank you for the valuable feedback and attention to detail. We address your concerns regarding the system design choice and performance below. The concerns about **Limited Action Space** and **Limited Broader Applicability** are also discussed in the [shared review](https://openreview.net/forum?id=lLq09gVoaTE&noteId=VK-BThqCDuD) above. Based on your comments, we have added Appendix D.1 to ablate HapticVisualNet's sample efficiency, made writing changes throughout for clarity, and uploaded the codebase and dataset to the website.
>
> **Concerns about the success rates and sample efficiency**
>
> We have since included a discussion on sample efficiency and model accuracy in Appendix D.1. Training HapticVisualNet on 300 examples yields an overall classification accuracy of 94% on held-out data, whereas models trained with less data achieve accuracies between 84-92%. We also note that although the raw data comprises 300 paired examples, we augment this dataset 8X with various color and affine transformations (see Appendix C). The discrepancy between 61-71% task success and 94% classification accuracy is due to the low-level challenges of manipulation rather than lack of training data. In particular, small inaccuracies with bounding box perception or visual servoing can compound, causing the fork to be misaligned with a food item during probing. Thus, even for a correctly classified primitive, the fork may cause damage, instability, or dropping. These challenges in precision are exacerbated in the case of items with irregular geometry or composition, such as broccoli. In the case of broccoli, the Haptic only baseline essentially acts as an “angled skewer-only” primitive. Haptic signals from probing broccoli at the head can be low in force magnitude, causing the network to mistakenly predict angled skewering. This ultimately fails to pierce the broccoli and achieves 41% success compared to 61% (OURS) with mostly vertical skewering. This demonstrates the benefits of selection between the primitives using our approach as opposed to relying on a fixed primitive.
>
> > After playing around with various foods, we think that both primitives could be combined into a vertical skewer motion and a subsequent angled lifting with support from the plate.
>
> Thank you for the observation, would you kindly mind clarifying what is meant by “angled lifting with support from the plate”?  In initial experimentation to determine a feasible action space, we started with only vertical skewering followed by angled lifting. We found that it works quite reliably for most items, with the exception of slippery and fragile items like banana slices and tofu. Even with highly dynamic and angled scooping afterwards, which may not be feasible in a safety-critical system, vertical skewering can cause dropping and slipping for delicate items which are not able to overcome gravity without angled support. Regarding ablations for the action space, the SPANet baseline is our main point of comparison. SPANet includes a larger taxonomy of 4 different primitives beyond the 2 which we consider, and we observe that it is biased towards vertical skewering in general. In [this video of SPANet](https://drive.google.com/file/d/1gBSZXYcP1v_P7r_wK07cKL3pyRpR9eIg/view), we observe consecutive failures with vertical strategies for banana slices (0:09 - 0:23), motivating the need for an angled approach, as seen in [this video of HapticVisualNet](https://drive.google.com/file/d/1HepjJ53aqbzC54KcConAxzyrSsbjMyNu/view) (0:23).
>
> **Clarifying the action space**
>
> We appreciate this suggestion! The action space itself does not make any assumptions about delta z or delta beta, but we have clarified the text in Section 4.2 regarding our assumptions about instantiating primitives within this action space.
>
> **Reward Definition**
>
> We only intend to formalize the bite acquisition setting as learning which primitive is most likely to succeed (essentially a sparse reward setting with 0 = fail, 1 = successfully skewered where a single time step denotes a single primitive attempt). You correctly note that this is baked into the binary classification loss HapticVisualNet is trained with.
>
> **Control stack and choice of force sensing**
>
> We found that the internal force/proprioceptive feedback we get from the Franka grippers is not fine-grained enough to sense deformable food item properties. Additionally, our instrumented fork design is an attachment to the existing Franka gripper hand, so using an external F/T sensor allows for obtaining force readings in the directions along the skewering axis. Thank you for the suggestion about including a discussion of the control software! We use a custom cartesian impedance controller implemented as a ROS 2 Python wrapper around libfranka (now updated in Section 4.2).

---

> > ### Author Response · Authors · 2022-08-22
> > **Response to Reviewer b6eX [2/2]**
> >
> > > Why is ServoNet necessary?
> >
> > The Franka only has strong precision guarantees within a very limited subspace of the entire robot workspace (40 x 40 x 40 cm. cube), so an arbitrarily placed plate of food can be out of this reach. It may definitely be possible to tune controller gains for better precision, but in practice we found this to require a lot of manual tuning which can be circumvented with learned servoing.
> >
> > Thank you for the catches about the typo and citations! We have revised the text and references accordingly. We thank you for your thorough review and constructive feedback, and are happy to address any other concerns or questions you may have. In the light of our additions, changes, and new experiments, we hope you consider raising your score.

---

> > > ### Author Response · Authors · 2022-08-25
> > > **Follow-Up to Reviewer b6eX**
> > >
> > > Thanks again for your feedback! We hope the response above has addressed your concerns, and we have since updated our response with additional physical experiments [here](https://openreview.net/forum?id=lLq09gVoaTE&noteId=nxY8TNT-ntO). With the rebuttal period ending soon, please let us know if you have decided to raise your rating, or if we can do anything to address any remaining thoughts.

---

### Official Review · Reviewer_LSMg · 2022-07-31

**Originality:** Very Good
**Technical Quality:** Very Good
**Clarity Of Presentation:** Excellent
**Impact:** 4

**Recommendation:**

Strong Accept: I recommend accepting the paper and will argue for my recommendation even if other reviewers hold a different opinion.

**Summary:**

This paper proposes a new bite-acqusition framework for robot skewering food on a plate using multimodal (visual and haptic) information. The framework compose of two stages: the probing stage and the skewering stage. During the probe stage, two networks are trained to localize the food bite and approach the localized food till contact is made. During the skewering stage, the proposed VisualHapticNet fuses the haptic measurements from a force-torque sensor and images from a camera to determine a skewering primitive (vertical versus angled). Comprehensive experimental results are conducted to test the framework's performance on in-distrituion and out-of-distribution food acquisitons, with comparisons to baselines and ablations. The results show good performance on food with in training distribution and adequate generalization to unseen food. Comparison to the basline and the ablations show the importance of using both modalities and using reactive planning.

**Issues:**

Please see the weakness section as above.

**Quality Of The Limitations Section:**

Limitations are addressed clearly

**Reviewer Expertise:**

4: The reviewer is confident but not absolutely certain that the evaluation is correct

**Robotics Focus:**

Sufficient demonstration on hardware

**Strengths And Weaknesses:**

Strength:
- Robot-assisted feeding is definitely an important application so the paper is well-motivated. The idea of combining visual and haptic information makes a lot of sense for this application.
- The experimental results are strong and pretty complete in my view. The system achieves good in-distribution and out-of-distribution performances . The failure cases are also shown which are appreciated. The comparison to baselines and abalations are well conducted and helped understand the contribution of each component of the method.
- The paper is clearly written and I enjoyed reading it.

Weakness:
I mostly have some minor comments:
- Can you provide more details regarding how the ServoNet is trained? Is the label just delta movement of the end-effector?
- What does '-' mean in Table 1? Does it mean 0 success or no experiment is conducted? If the latter, why?




**Summary Of Recommendation:**

I think the proposed idea of combining visual and haptic information for robot assisted feeding makes a lot of sense, and the experimental results are impressive and strong and supports the proposed idea.

---

> ### Author Response · Authors · 2022-08-22
> **Response to Reviewer LSMg**
>
> Thank you for your comments, and we are glad to hear that you enjoyed the read! Below we address your clarification questions. Based on your comments, we have added Appendix C which includes additional details about ServoNet.
>
> > Can you provide more details regarding how the ServoNet is trained? Is the label just delta movement of the end-effector?
>
> Thank you for the question and apologies for the lack of clarity! The dataset consists of paired plate images and annotations of the fork keypoint and nearest food item keypoint as pixels. We train a ResNet-18 FCN to regress these keypoints from visual input. At test time, we use the offset in predicted fork position and food position to guide the robot closer to the food item until the keypoints align within some fixed threshold. We have updated the main text (Section 3.5) and supplementary material (Appendix C) to include a longer discussion of training and implementation details.
>
> > What does '-' mean in Table 1? Does it mean 0 success or no experiment is conducted? If the latter, why?
>
> As Reviewer jNun also pointed out, this indicates that no experiment was conducted as the specific method is not applicable in the testing scenario. We kindly refer to [our response above](https://openreview.net/forum?id=lLq09gVoaTE&noteId=0P5VvDIyG2) for additional explanation.

---

### Official Review · Reviewer_NczR · 2022-08-01

**Originality:** Very Good
**Technical Quality:** Excellent
**Clarity Of Presentation:** Excellent
**Impact:** 3

**Recommendation:**

Strong Accept: I recommend accepting the paper and will argue for my recommendation even if other reviewers hold a different opinion.

**Summary:**

In the paper, the authors introduce a new network that incorporates both haptic and visual data in order to determine the best fork skewering skill parameters after first probing the item. It differs from previous work that relied solely on visual data to determine fork skewering locations. They ran many studies on both unseen and seen food on 6 different plates of food items where sometimes the food items were of the same type but some of the food was boiled so they were visually similar but had vastly differing properties. Their network improved on previous skewering networks and they tested the success results of the different ablations on the real robot.

**Issues:**

I don't really have any issues with the paper. It is well written and the video is excellent as well. Overall, if you could address maybe some of the issues I brought up in the strengths and weaknesses section above that would be ok. For example, maybe doing some more experiments on frozen fruit vs normal fruit or sauteed / wok fried objects with sauces vs normal items. In addition, the food items on the plates were very separated. If there was a normal served plate of food where lets say rice or mashed potatoes is in one corner, the vegetables are in another corner, and there are some stir fried chicken with bell peppers or something, how would the Haptic Visual Net react to more cluttered plates and what new data would be necessary to increase the robustness of your method so that it would scale for real-world adoption?

**Quality Of The Limitations Section:**

Limitations are addressed clearly

**Reviewer Expertise:**

5: The reviewer is absolutely certain that the evaluation is correct and very familiar with the relevant literature

**Robotics Focus:**

Sufficient demonstration on hardware

**Strengths And Weaknesses:**

The main strengths of the paper is that they were able to train this Haptic Visual Net using only human annotated data that required only 3 hours of data collection and labeling and didn't need to waste time to have the robot try actual skewering attempts and then a human would have to reset a scene and then clean the fork. That was actually very surprising to me. The original SPANet required 81 hours of human labeling or robot data collection according to their paper. While I have not read that paper, it seems credible in my experience of doing my own robot experiments. I also believe that sometimes if a human can give sufficient labels that are accurate enough, then trying to do the experiments with an actual robot will take more time.

I only wish that the authors tried a couple more different varieties of hard and soft objects of the same category. For example, frozen fruits vs regular room temperature fruits. They seemed to only do boiled carrots. The boiled carrots and pasta as unseen was very interesting. Maybe for future work, they can try running the network on cooked items like sauteed chicken with vegetables in more cluttered plates than the much less dense plates that were used for evaluation. If the robot were to skewer multiple thing at once because they were overlapped, what would Haptic Visual Net do in case some items are not skewered well enough and they were to fall on a person and unfortunately hurt them if they were hot? Also what would be the effect of sauces be on networks like if they were charred (soft) vs. dark soy sauce on the objects. Would they be classified similarly? I hope that people that might need the robot don't just have to eat plain vegetables without any sauce haha. Also as food items cool, they might change as well so maybe there should be in between classes instead of soft and hard. For example, chewy or sticky or something.

Finally, I wonder if sometime it would be more useful if the robot could also use a bandit formulation of exploration vs exploitation to figure out whether all the food items on the plate are similar and whether to continue probing or whether it is making a lot of mistakes skewering to then probe the items more to figure out its properties. I wouldn't want a robot to waste time continuously probing if every item was as predicted.


**Summary Of Recommendation:**

Overall, I give this paper a strong accept. I think the well-written paper and accompanying video can be useful for the robotics community. While the problem of fork skewering is not novel, they were able to develop a new network using a probing action and the haptic feedback from the action instead of only relying on vision to determine the parameters of the fork skewering actions. I see no reasons why someone would reject the paper and would argue for my recommendation. I felt like the experiments were well executed and the results are in line with what I would hope for a personal assistive feeding robot. However, there could be more experiments with food items that are the same type but different properties like frozen fruits / vegetables or slightly sauteed items vs regular food items.

---

> ### Author Response · Authors · 2022-08-22
> **Response to Reviewer NczR**
>
> We thank the reviewer for your thoughtful feedback and appreciate hearing that you enjoyed the read! It seems that two main areas for improvement are understanding HapticVisualNet’s generalization capabilities and how the method would scale to more complex scenarios, which we cover below and run additional experiments for (Appendix E).
>
> > There could be more experiments with food items that are the same type but different properties like frozen fruits / vegetables or slightly sauteed items vs regular food items.
>
> Thank you for this suggestion! We have evaluated HapticVisualNet on physical rollouts with frozen fruits (pineapple, mango, and strawberry) and observe a 20/25 success rate, with most failures attributed to slipping on highly rigid frozen items during probing. HapticVisualNet appropriately infers vertical skewering for most attempts as expected with items of such rigidity, but occasionally predicts angled skewers which we hypothesize is due to thawing. We summarize the failure modes and provide videos of these rollouts in Appendix E. We are currently in the process of running experiments with sauce-coated veggies and tofu. We will update the manuscript and response by the end of the week with more information on the results of these experiments.
>
> > If the robot were to skewer multiple things at once because they were overlapped, what would Haptic Visual Net do in case some items are not skewered well enough?
>
> This is a great point and a strong motivation for using ServoNet to precisely approach an isolated item without skewering multiple things at once. We found this to be crucial to avoiding multiple skewers within the same attempt. In future work, we hope to incorporate visual feedback to detect overlapping items, especially on more cluttered plates. This could inform a sequence of skewering actions which attempts the least occluded items first.
>
> > I wonder if sometime it would be more useful if the robot could also use a bandit formulation of exploration vs exploitation to figure out whether all the food items on the plate are similar and whether to continue probing or whether it is making a lot of mistakes skewering to then probe the items more to figure out its properties.
>
> This is another great suggestion! Prior work uses a contextual bandits formulation where the context is an image observation of the food item, the arms represent different primitives, and the post-hoc haptic context observed after an attempt is used to update action likelihoods [2]. While relying on post-hoc context over multiple attempts could damage an item, our approach (HapticVisualNet) instead uses visual and haptic info within the same interaction to plan reactive skewering. In the future, it would certainly be interesting to only probe items that are unfamiliar and otherwise extrapolate previously successful strategies to familiar items.
>
> > How would the Haptic Visual Net react to more cluttered plates and what new data would be necessary to increase the robustness of your method so that it would scale for real-world adoption?
>
> Likely, training the individual system components (bounding box detector, ServoNet, HapticVisualNet) on a wider distribution of food items would be one step closer towards real-world adoption. It would also be interesting to automatically detect when a skewering action succeeds/fails, and use this to do self-supervised dataset collection at scale for paired images/haptic readings/successful primitive labels. The action space may also need to be enlarged. We may want to consider the item surface geometry when planning the angle of fork insertion for example, instead of assuming top-down insertion. Cluttered also introduces a higher-level question of inferring the order in which to pick items, which may require sensing occlusion or the stable poses of an item.

---

> > ### Comment · Reviewer_NczR · 2022-08-25
> > **Response to Author Rebuttal**
> >
> > Thank you for taking the time to conduct the additional skewering experiments on frozen and sauteed foods. I await the results in the final version of the paper during the rebuttal phase, but I will most likely not be adjusting my ratings due to my already strong accept rating.

---

> > > ### Author Response · Authors · 2022-08-25
> > > **Follow-Up on Additional Experiments**
> > >
> > > Thank you! We very much appreciate your rating, and have updated [our response above](https://openreview.net/forum?id=lLq09gVoaTE&noteId=nxY8TNT-ntO) with the experiments on sauteed foods.

---

### Official Review · Reviewer_jNun · 2022-08-01

**Originality:** Good
**Technical Quality:** Good
**Clarity Of Presentation:** Good
**Impact:** 3

**Recommendation:**

Weak Accept: I recommend accepting the paper, but will not argue for my recommendation if the majority of other reviewers have a different opinion.

**Summary:**

This paper presents a system for food skewering by considering both visual and haptic information. Specifically, the authors propose a neural network-based framework that fuses the multimodal information during a single skewering interaction to learn food representations (via supervised learning), informing the food item's geometry and deformation properties. The model is learned by mapping visual and haptic information to a discrete set of skewering primitives using a small manually-labeled dataset collected in an offline fashion. The resulting system can then be used to skewer food items.

The authors have evaluated the proposed system on a diversified set of food items with varying geometry and deformability, achieving 71% skewering success across 21 items total. The experiment results also suggest that the consideration of multiple sensory modalities enables the proposed method to handle unseen objects and outperform those that only take single modality input.

**Issues:**

I have concerns regarding
- the limited action space,
- details about the reactive/feedback process,
- more comprehensive evaluations of the amount/distribution/diversity/ambiguity of the training data, and
- certain claims about the contribution of this paper.

Please see the weaknesses section for more details.

In addition, why are there missing numbers in Table 1 (slots marked with "-")? Is it because the specific methods are not applicable in the testing scenario, or did the authors not get the chance to run the experiments?

**Quality Of The Limitations Section:**

Additional details required

**Reviewer Expertise:**

4: The reviewer is confident but not absolutely certain that the evaluation is correct

**Robotics Focus:**

Sufficient demonstration on hardware

**Strengths And Weaknesses:**

**[Strengths]**

This paper tackles a very interesting problem of food skewering. The task is challenging as the robot has to deal with objects with different geometry and complicated physical properties, but this is also a necessary step for robot-assisted feeding. A satisfying solution to this task holds the potential to achieve great real-world impact of having robots caring for our elderly population or people with disabilities.

The authors have evaluated the proposed system on a diversified set of objects of different visual appearances and deformability. They have also provided both quantitative and qualitative evaluations demonstrating the effectiveness of the proposed method. The video demonstrations in the supplementary materials and the website are especially helpful for the readers to understand the difficulty of the task and the system's performance.

This paper also provides many very practical treatments for the design of different system components and discusses why the task is hard with concrete examples (e.g., distinguishing between raw and boiled carrots). A lot of the observations obtained through the experiments and discussions of the failure modes are also very interesting and tremendously helpful for the follow-up work to make further progress.

**[Weaknesses]**

My primary concern with this work is that the action space is a bit limited. The authors mainly consider two types of pre-specified action primitives designed specifically for this task: angled skewering and vertical skewering. I understand that these two primitives may be enough for the food skewering tasks, but such constraint limits the method's applicability in other tasks. It is also hard to know how well the method can scale as the number of action primitives increases.

The authors also claim that the system learns a reactive policy. I thus hope the authors can explain in more detail how the feedback is taken during the task execution process and the frequency of the feedback loop. Correct me if I am wrong, but it seems that the robot does the task in a two-step process: probe, then skewer, where both steps are executed in an open-loop fashion. Some of the videos also show that even if the robot fails to skewer a food item, the robot continues to perform the scoop motion, indicating that no feedback is considered during this stage. Will feedback at a higher frequency help here? I'm curious to know the authors' perspectives on this and how to incorporate such visuomotor closed-loop ability in the system.

One thing missing from the experiments is a more comprehensive evaluation of how much data is needed and how the data's size and diversity influence the performance and generalization abilities. Such experiments can be essential to understanding the sample efficiency and the applicability of the method.

Meanwhile, for generalization to unseen objects, are there ways to quantify the distance between the unseen objects and the training distribution (e.g., according to the shape and stiffness)? It would be helpful for the readers to understand to which extent can the learned system generalize and how the system behaves as it encounters less and less familiar objects.

I could also imagine that the labeling of the objects can be ambiguous. For example, some food items with deformability between the banana and the broccoli may be successfully skewered via either angled or vertical skewering actions. Do the ambiguities in labeling matter during the learning process, and how do such ambiguities be resolved in the pipeline? Will a more continuous action representation help in this case?

In Figure 4, Cumulative Failures, why does HapticVis (the proposed method) perform worse than Haptic and Vis alone in BBox FP?

The first contribution stated in the introduction discussed a coarse-to-fine visual servoing method to approach a food item. However, no mention of it is found in the remainder of this paper. If such a coarse-to-fine visual servoing method is not the main focus but a contribution of another paper, I suggest the authors rephrase the text to highlight the unique contribution made by this paper.

**Summary Of Recommendation:**

I think this is a well-executed paper tackling an important problem of food skewering. The idea of fusing multiple sensory modalities (i.e., haptic and vision) is also interesting and has shown to be effective in this specific task. The authors have also provided extensive evaluations in the real world with food items of diverse visual appearance and physical properties. I thus lean towards the acceptance side.

---

> ### Author Response · Authors · 2022-08-22
> **Response to Reviewer jNun**
>
> Thank you for the detailed review! We have addressed some of your concerns in the shared response above, specifically regarding **Limited Action Space** and **Limited Broader Applicability.** Based on your suggestions, we have also updated Appendix C with additional training details about ServoNet and Appendix D.2 to interpret learned HapticVisual features.
>
> > Explain in more detail how the feedback is taken during the task execution process.
>
> The policy is reactive at the level of primitive selection (choosing a primitive at the point of contact with an item), but not fully reactive in a visuomotor sense as primitives are executed in an open loop fashion.
>
> > Are there ways to quantify the distance between the unseen objects and the training distribution (e.g., according to the shape and stiffness)?
>
> This is an interesting question! Quantifying this is an open challenge even in the more general field of multitask and transfer learning, and as you mention there are many possible reasons for an item lying out of distribution. However, we agree that assessing the patterns that emerge for learned features of items with similar/dissimilar properties would improve the interpretability of HapticvisualNet. To that end, we have updated the paper (Appendix D.2) with tSNE projections visualizing the learned multimodal food embeddings by training item class. The learned multimodal embedding space clusters items according to 1) shared visual/textural properties and 2) favorable manipulation strategies as desired. The distance metric based on this embedding space can help with assessing in- and out-of-distribution training and transfer.
>
> > Some food items may be successfully skewered via either angled or vertical skewering actions. Do the ambiguities in labeling matter? Will a more continuous action representation help in this case?
>
> Thank you for this question, we certainly agree that there exists a spectrum of deformability for a given food item that a discrete action space cannot fully describe. However, initial experimentation showed that small changes in the angle of fork insertion do not make a perceptible impact on skewering success as much as being able to recognize whether an item is delicate or rigid. These initial experiments for recognizing when vertical or angled skewering is appropriate, as well as human priors over when a food is compliant vs. not, were essential to labeling consistently. To resolve ambiguity at test time, we always execute the maximum likelihood primitive, although in the future it would be interesting to use these likelihoods to quantify uncertainty. We also note that there are many other haptic adjectives to describe food beyond softness/hardness (stickiness, compliance, elasticity, etc.) [3] that indeed a continuous action representation may capture better.
>
> > In Figure 4, Cumulative Failures, why does HapticVis perform worse than Haptic and Vis alone in BBox FP?
>
> BBox FP is only a failure mode of high-level object detection and is unrelated to the low-level skewering policy (HapticVis, Haptic, Vis, SPANet) failures, so this difference is just attributed to randomness across plates.
>
> > The first contribution stated in the introduction discussed a coarse-to-fine visual servoing method to approach a food item. However, no mention of it is found in the remainder of this paper. I suggest the authors rephrase the text to highlight the unique contribution made by this paper.
>
> Thank you for pointing this out. ServoNet is a contribution of this paper, and is more of a system-level contribution that  we found critical to ensuring robust skewering. We have expanded on the training and implementation details in the supplementary material (Appendix C).
>
> > Why are there missing numbers in Table 1 (slots marked with "-")?
>
> We apologize if “-” was unclear. We have now added a note in the table caption clarifying the meaning. These are experiments that do no not yield useful comparisons. We would like to emphasize that after conducting all methods on Plates 1-3, we wanted to focus our comparisons on the most relevant baselines in the remaining testing scenarios. The broccoli experiment (Plate 4) is an ablation to test whether haptic information alone or joint visual and haptic cues are necessary for items of irregular shape and composition, since broccoli yields very different haptic signals when probed at the head (low force at leafy region) vs. the stem (high force at stem). We hypothesize that this can be disambiguated with vision. Thus we compare HapticVis and Haptic for this plate, and find that Haptic alone wrongly predicts angled skewering especially when probing at the head, failing to pierce the item, whereas HapticVis correctly infers vertical skewering most of the time. For Plates 5 and 6, we mainly are interested in whether jointly fusing haptics and vision is necessary for OOD plates, compared to vision alone as in prior work [1]. Thus we compare against Vis only for these plates.

---

> > ### Author Response · Authors · 2022-08-25
> > **Follow-Up to Reviewer jNun**
> >
> > We hope we have addressed your concerns, and very much appreciate the feedback! With the rebuttal period coming to a close, please let us know if you would like to see anything else on our end.

---

> > > ### Comment · Reviewer_jNun · 2022-08-28
> > > **Thank you for the feedback!**
> > >
> > > Thank you for the detailed feedback and the additional experimental results! They addressed the majority of my concerns. I keep my score as Weak Accept.

---

### Meta-Review · Area_Chair_48a5 · 2022-08-14

**Recommendation:** Accept (Oral)
**Confidence:** 4

**Metareview:**

The paper proposes a system for food skewering by considering both visual and haptic information and evaluates it thoroughly,

Strengths:
- Interesting problem of food skewering (jNun, NczR)
- relatively little data required (NczR)
- good evaluation (jNun, NczR)

Weaknesses:
- limited action space (jNun)
- limited amount of hardnesses for the same objects (NczR)
- ablations on the amount of data missing (b6eX)

This is just an excerpt of the reviews. I invite the authors to respond to the reviewers and address their concerns.

Post-rebuttal:
The authors have been able to address many of the concerns and the paper should definitely be accepted. Since the paper is round and shows great results on a real robot it might be considered for an oral (weak endorsement).

---

> ### Author Response · Authors · 2022-08-22
> **Response to Area Chair 48a5**
>
> Thank you to everyone for your valuable time and feedback! We are glad to hear that the reviewers agreed that our framework for reactive, multimodal, zero-shot bite acquisition is data-efficient and thoroughly evaluated.
>
> For brevity, we address shared concerns here, and address each reviewer’s comments point-by-point in the sections below:
>
> **Limited Action Space (Reviewer jNun, Reviewer b6eX)**
>
> The reviewers are concerned about the limited action space of our primitives. While we agree that our algorithm with the two classes of action primitives does not encompass all possible food, we ourselves were surprised by the wide range of food items that could be skewered using simply two primitives. Our two primitives are very much capable of modulating the fork insertion angle upon contact. Specifically, the primitives are expressive enough to handle the same if not wider range of food items than prior work (that include a larger taxonomy of six primitives [1]). In addition, by design, our small action space enables tractable supervised learning. In the future, it would be interesting to expand the set of food items even more and that could potentially require a continuous action space. We believe our discretized primitives could be useful even in that setting to bootstrap self-supervised exploration in a continuous action space.
>
> **Limited object properties considered (Reviewer jNun, NczR)**
>
> To supplement the existing experiments which test boiled, raw, and microwaved varieties of food items, we have since run additional experiments with HapticVisualNet on frozen fruits as suggested by Reviewer 2 and are in the process of completing rollouts on sauteed foods with sauce to stress-test the generalization capabilities. We will report back on the results of these experiments by the end of the rebuttal period.
>
> **Ablations on diversity and amount of data missing (Reviewer LSMg, b6eX)**
>
> As suggested by Reviewers 3 and 4, we have added additional experiments that validate the sample efficiency and quality of learned HapticVisualNet representations across diverse foods via ablations on amount of training data (Appendix D.1) and t-SNE visualizations (Appendix D.2), respectively.
>
> **Limited broader applicability (Reviewer jNun, NczR)**
>
> While our primitives are specific to food skewering, we hope that our general framework for reactive visuo-haptic planning can be applicable in other contact-rich domains such as peg transfer [4], handovers [5], and grasping [6] in the future.  In our setting, the action space is small allowing for supervised learning to train a policy. In a more continuous action space, it may be possible to instead collect demonstrations of a desired task resulting in paired images, haptic readings, and action taken with which to train a visuo-haptic policy.  In addition, we would like to emphasize that our work develops a system that robustly skewers a wide range of food items. We believe this system is quite general, impacting the field of assistive feeding.
>
> **We have attached a new version of the manuscript with changes highlighted in teal. Our specific changes based on the reviewers’ feedback include:**
> * Added codebase + dataset to the [website](https://sites.google.com/view/hapticvisualnet-corl22)
> * New physical experiments on 1) frozen fruits and ongoing experiments with 2) sauce-coated tofu and vegetables (See Appendix E)
> * New ablations on sample efficiency (See Appendix D.1)
> * Additional details about ServoNet (See Appendix C)
> * Interpretable visualization of HapticVisualNet learned representation (See Appendix D.2)
> * Clarifications in the main text including revising the action space definition, writing changes throughout, and replacing arXiv’d citations with their conference/journal counterparts
>
> **References**
>
> [1] *Feng, Ryan, et al. "Robot-Assisted Feeding: Generalizing Skewering Strategies Across Food Items on a Plate." The International Symposium of Robotics Research. Springer, Cham, 2019.*
>
> [2] *Gordon, Ethan K., et al. "Leveraging Post Hoc Context for Faster Learning in Bandit Settings with Applications in Robot-Assisted Feeding." 2021 IEEE International Conference on Robotics and Automation (ICRA). IEEE, 2021.*
>
> [3] *Gemici, Mevlana C., and Ashutosh Saxena. "Learning haptic representation for manipulating deformable food objects." 2014 IEEE/RSJ International Conference on Intelligent Robots and Systems. IEEE, 2014.*
>
> [4] *Lee, Michelle A., et al. "Making sense of vision and touch: Self-supervised learning of multimodal representations for contact-rich tasks." 2019 International Conference on Robotics and Automation (ICRA). IEEE, 2019.*
>
> [5] *Pan, Matthew KXJ, Elizabeth A. Croft, and Günter Niemeyer. "Exploration of geometry and forces occurring within human-to-robot handovers." 2018 IEEE Haptics Symposium (HAPTICS). IEEE, 2018.*
>
> [6] *Murali, Adithyavairavan, et al. "Learning to grasp without seeing." International Symposium on Experimental Robotics. Springer, Cham, 2018.*

---

> > ### Author Response · Authors · 2022-08-25
> > **Additional Experiments Update**
> >
> > **UPDATE:**
> >
> > Per the reviewer's suggestions, we have since ran additional physical experiments on both frozen fruit and sautéed/stir-fried vegetables and tofu to stress-test the capabilities of HapticVisualNet, observing the following results:
> >
> > | Items | Skewering Success Rate | Video | Slip/Miss | Bounding Box False Negatives | 3+ Tries
> > | ----------- | ----------- | ----------- | ----------- | ----------- | ----------- |
> > | Frozen mango, pineapple, strawberry | 20/25 (80%) | [[Video]](https://drive.google.com/file/d/1Cvv4XKfGlyfuzuAI_GgpIpFUbtcVX0Us/view) | 5 | 0 | 1
> > Sauteed veggies and tofu | 20/23 (87%) | [[Video]](https://drive.google.com/file/d/1dWa7HlcuBp_V45Z7TJBb5xdMgpgKw6-A/view?usp=sharing) | 3 | 3 | 0
> > Sauteed veggies, tofu, soy sauce | 25/34 (74%) | [[Video]](https://drive.google.com/file/d/1dWa7HlcuBp_V45Z7TJBb5xdMgpgKw6-A/view?usp=sharing) | 9 | 6 | 1
> > |  **Overall** | 65/82 (79%) |  |  |  |  |
> >
> > The attached updated paper draft reflects these results and an analysis of performance/failure modes.